# Evaluation of a New, Highly Flexible Radiofrequency Coil for MR Simulation of Patients Undergoing External Beam Radiation Therapy

**DOI:** 10.3390/jcm11205984

**Published:** 2022-10-11

**Authors:** Kiaran P. McGee, Norbert G. Campeau, Robert J. Witte, Philip J. Rossman, Jackie A. Christopherson, Erik J. Tryggestad, Debra H. Brinkmann, Daniel J. Ma, Sean S. Park, Dan W. Rettmann, Fraser J. Robb

**Affiliations:** 1Department of Radiology, Mayo Clinic and Foundation, Rochester, MN 55905, USA; 2Department of Radiation Oncology, Mayo Clinic and Foundation, Rochester, MN 55905, USA; 3GE Healthcare, Waukesha, WI 53188, USA

**Keywords:** radiofrequency coils, treatment simulation, MRI, head and neck, adaptive image receive (AIR) coil

## Abstract

Purpose: To evaluate the performance of a new, highly flexible radiofrequency (RF) coil system for imaging patients undergoing MR simulation. Methods: Volumetric phantom and in vivo images were acquired with a commercially available and prototype RF coil set. Phantom evaluation was performed using a silicone-filled humanoid phantom of the head and shoulders. In vivo assessment was performed in five healthy and six patient subjects. Phantom data included T_1_-weighted volumetric imaging, while in vivo acquisitions included both T_1_- and T_2_-weighted volumetric imaging. Signal to noise ratio (SNR) and uniformity metrics were calculated in the phantom data, while SNR values were calculated in vivo. Statistical significance was tested by means of a non-parametric analysis of variance test. Results: At a threshold of *p* = 0.05, differences in measured SNR distributions within the entire phantom volume were statistically different in two of the three paired coil set comparisons. Differences in per slice average SNR between the two coil sets were all statistically significant, as well as differences in per slice image uniformity. For patients, SNRs within the entire imaging volume were statistically significantly different in four of the nine comparisons and seven of the nine comparisons performed on the per slice average SNR values. For healthy subjects, SNRs within the entire imaging volume were statistically significantly different in seven of the nine comparisons and eight of the nine comparisons when per slice average SNR was tested. Conclusions: Phantom and in vivo results demonstrate that image quality obtained from the novel flexible RF coil set was similar or improved over the conventional coil system. The results also demonstrate that image quality is impacted by the specific coil configurations used for imaging and should be matched appropriately to the anatomic site imaged to ensure optimal and reproducible image quality.

## 1. Introduction

Radiation oncologists have long recognized the value of MR imaging in RT treatment simulation [1,2,3,4,5,6]—in particular, the ability to generate multiple soft tissue contrasts, as well as the quantitative assessment of various functional parameters that are known indices of disease type, stage, and response to therapy, such as perfusion, diffusion, tissue mechanical properties, and blood oxygenation. These biomarkers hold the promise of allowing the more precise delineation and characterization of tumor volumes and associated organs at risk (OAR), thereby minimizing treatment margins, potentially reducing side effects and improving outcomes [6,7,8]. With the development of ultra-short and zero echo time (uTE, ZTE) [9,10], atlas matching [11,12,13,14], and deep learning algorithms [15] to generate synthetic or pseudo CT (pCT) data, the last hurdle to moving towards MR-only RT treatment simulation would seem to have been overcome.

In practice, a limitation of MR RT treatment simulation, i.e., the acquisition and use of MR imaging data to design and optimize radiation therapy treatments, has been the ability to obtain high-quality and reproducible MR images in treatment position. A major contributor is the limited number of dedicated RF coils that can be adapted to provide sufficiently high image quality both in terms of SNR and uniformity [16]. There are several reasons for this. First, MR scanner manufacturers have designed dedicated RF coils to provide the highest-quality images by creating fixed coil geometries encased within tightly conforming rigid housings. These housings serve to protect the fragile coil elements from damage due to mechanical forces and ensure a fixed coil geometry, as well as bringing the individual coil elements that make up the coil as close to the patient as possible. Second, the use of rigid, small-volume RF coils means that patients are required to be imaged in a neutral position. While this has served the diagnostic radiology community well, it has hindered the imaging of RT patients, who are often imaged in their treatment position and immobilization device. Based on our own clinical experience, this is particularly evident in the imaging of the brain, head and neck, and lower cervical/upper thoracic spine. Finally, while manufacturers do provide RF surface coil arrays that can be placed directly onto a patient, they do not provide the flexibility needed to follow the exact external contours of the patient in treatment position, are challenging to place, and result in inhomogeneous signal intensities that can only be partially corrected using post-processing techniques such as signal intensity correction and denoising. Of these, the brain represents the most amenable site for the placement of smaller flexible arrays due to its roughly spherical shape, but still presents with unique challenges in terms of reproducible image quality due to their semi-rigid construction and overall thickness. What is needed is a lightweight, highly flexible RF coil that can closely follow the surface contours of patients being imaged in their RT treatment position over a range of body habitus and immobilization devices while ensuring optimal SNR and signal uniformity.

Recent developments in RF coil design and engineering have resulted in the introduction of extremely lightweight and flexible coil systems that are ideally suited to address the limitations imposed by existing RF coil technologies, particularly in the context of MR treatment simulation. Previous investigations [17] have demonstrated that this technology does not require fixed spacings between individual coil elements necessary to reduce mutual inductance between adjacent elements, thereby increasing the number of coils that can be placed within a given geometry (i.e., increased RF coil density), are highly flexible due to the elimination of lumped coil components such as capacitors, and are lightweight due to the use of thin and extremely flexible conducting loops instead of copper conductors printed onto a semiflexible substrate such as fiberglass [18]. Additionally, they are particularly well suited to the challenges of MR simulation of the head and neck, which requires highly flexible coils to conform to the complex surface contours of this anatomic region, high SNR to minimize susceptibility induced signal loss and distortion, particularly at high field strengths (i.e., 3T), and enable high resolution images to precisely visualize the target volume and adjacent OAR. 

We hypothesized that a novel prototype coil system that uses a recently introduced RF coil technology known as the adaptive image receive (AIR) [17,19] coil system would be ideally suited for MR in treatment position of head and neck RT simulations compared to traditional RF coil systems. The purpose of this study was to test this hypothesis in both phantoms and humans by comparing the performance of a traditional, commercially available RF coil system designed for MR simulation of head and neck patients to a prototype AIR coil system. 

## 2. Methods and Materials

### 2.1. RF Coil Configurations

Two RF coil sets were evaluated. The first consisted of the manufacturer’s US Food and Drug Administration (FDA) 510(K) cleared system, as described in Table A1 of Appendix A and labelled as the RT SUITE coil set (GE Healthcare, Waukesha, WI, USA). This coil set used conventional copper-based RF coil loops integrated into rigid and semi-rigid housings, including a posterior array (RT open array), a rectangular flexible array (license plate), and two flexible ‘paddle’ arrays. The posterior array comprised eight individual elements arranged in an inverted T shape, in which four elements covered the most inferior portion of the coil left to right and the remaining four were perpendicular and superior to the inferior set. The rigid surface of the coil lies flush with the top surface of the MR table at the location of the RF head coil. The rectangular array consists of 16 coil elements arranged in a semiflexible housing, while the paddle arrays included three elements per paddle. Figure 1b,d show the placement of the separate coil components onto the phantom, which is the configuration used for routine clinical imaging. This does not represent the placement recommended by the manufacturer, which consists of using a custom coil positioning device in which each paddle array is placed along both sides of the subject’s head and the rectangular array suspended above the patient’s clavicles with the coil’s longest dimension left to right. Instead, the positioning used in this work is an adaptation based on in-house testing and optimization. The second consists of a prototype two-component RF coil set (NeoCoil LLC, Pewaukee, WI, USA) that was constructed using a new type of RF coil element that has been previously described and characterized [19]; it is shown in Figure 1c,e and referred to as the RT AIR coil. Unlike conventional copper RF coil elements, RT AIR coil elements are constructed of a lightweight, highly flexible continuous thin wire loop interfaced to a high-impedance digitizer and amplifier circuit. The low mass and flexible characteristics of these elements, as well as the low mutual inductance, mean that a light, extremely flexible, and high-coil-density system can be achieved compared to conventional RF coils. This coil consisted of a 15-element face array and seven-element anterior array designed to provide left–right coverage of the chest and shoulders, as seen in Figure 1c,e. Because the RT AIR coil configuration only included components designed to cover the head and anterior chest, the RT AIR coil was combined with the RT open array in software by means of a configuration file (i.e., coil configuration) to provide posterior signal coverage. As such, the coil represents a hybrid rather than an AIR coil-only system. In addition, three separate coil configurations were provided by the coil manufacturer, as described in Table A1. The FACE_RT configuration used only elements within the AIR face and posterior RT open array, while the FACE_AA_RT used all elements of the AIR coil in combination with the four inferior elements of the posterior RT open array. The final configuration—FACE_RT_AA_PA—included combining all elements of the RT AIR coil, the inferior four elements of the posterior RT open array, and two superior elements of the posterior spine array that is imbedded into the MR table. Common to all coil sets and coil configurations was the method in which individual coil signals were combined to produce the final composite image and involved calculation of the square root of the sum of the squared signal from each coil element activated by the individual coil configuration file. Combination of individual coil images was thus independent of the type of coil element. 

### 2.2. Phantom Data

A humanoid phantom was constructed by filling a plastic mannequin male head form (Model No. DMGYR, Zing Display, Rancho Santa Margarita, CA, USA) with 50% polyvinyl chloride (PVC) (#502 Plastic, Lure Parts Online, Inc., Springfield, IL, USA), 50% softener (#6705 Plastic Softener, Lure Parts Online, Inc., Springfield, IL, USA), and two rectangular, dome-shaped forms to mimic the shoulders. The PVC/softener combination produced a soft, solid material that did not evaporate over time or result in ‘swirling’ imaging artifacts due to motion seen when fluids are used as signal generating materials. The internal volume of the male phantom was 6.65 L, and 2.22 L for each shoulder form. The measured T_1_ and T_2_ relaxometry values of the phantom were 225 and 31 msec, which were estimated using inversion recovery and multiple echo time (TE) spin echo pulse sequences, respectively [20]. 

The phantom configuration was imaged on a 70 cm bore diameter 3.0T MR system (750W, GE Healthcare, Waukesha, WI, USA) used for routine MR imaging of RT patients. The phantom was placed on top of a universal MR-compatible couch top (CIVCO Radiotherapy, Coralville, Iowa, USA) and positioned so that the top of the phantom was parallel to the superior edge of the top two RF coils in the posterior RT open array. Images were acquired using a T_1_-weighted 3D variable flip angle multi-echo spin echo pulse sequence (CUBE, GE Healthcare, Waukesha, WI, USA) with the following scan parameters: field of view = 44 cm, pulse repetition rate/echo time = 650/11.9 msec, echo train length = 24, bandwidth = 125 (±62.5) kHz, slice thickness = 1.6 mm, acquisition matrix = 256 × 224 × 160 (frequency × phase × slice), k-space acceleration factors = 2 × 2 (phase × slice), 3D distortion correction on. Sequential acquisitions were obtained to reconstruct SNR data in accordance with the method proposed by the National Electrical Manufacturers Association (NEMA) standards, Publication MS 6-2008 [21], and described in the following paragraph. Imaging was performed using the RT SUITE configuration, followed by imaging with the RT AIR coil system. For the RT AIR coil set, acquisitions were repeated for each of the three separate coil configurations. Figure 1 shows the phantom and two RF coil sets (RT SUITE—Figure 1b,d and RT AIR—Figure 1c,e).

In compliance with the recommendations provided in the report on MRI simulation in radiation therapy published by the American Association of Physicists in Medicine (AAPM) task group (TG) 284 (see Table VI of the report) [16], quantitative assessment of phantom image quality involved calculation of both SNR and image uniformity metrics. An SNR volume was calculated by first measuring the standard deviation (SD) of the difference in the sequential T_1_-weighted 3D CUBE acquisitions described previously. The SD value was calculated over a region of interest within the difference (i.e., subtracted) volume equal to 40 × 170 × 10 (X × Y × Z) pixels centered at the reference position (center of phantom and anatomic landmark), in accordance with the recommendations provided by the NEMA-recommended SNR measurement protocols [21,22]. This value was then divided into the first of the two T_1_-weighted 3D CUBE volumes to create a 3D SNR map. The SNR maps were not scaled by 0.655 ($\sqrt{2}$) [21] as all data were evaluated as paired data sets and therefore represented a common scaling factor, nor was the SD of the noise divided by the coil scaling factor to convert the value to the equivalent Gaussian noise statistic recommended by the NEMA MS 9-2008 protocol [22], since this was also constant and equal to 0.71 for all coil types and configurations. After calculation of the 3D SNR maps, the volume was thresholded to only include those pixels within the phantom. A histogram of all SNR values within the phantom, as well as a per slice average SNR value, was then calculated. A total of 250 bins were used to generate the histogram of the entire (i.e., global) SNR data. The per slice average SNR was estimated by averaging the SNR values within the phantom for each slice. A second image quality metric, referred to as the per slice uniformity, was estimated and involved calculating the ratio of all SNR values within a given slice that were within ± 20% of the mean SNR value within the entire phantom divided by the number of pixels within the phantom for the slice of interest. This value ranged between 0 and 1.0, where 1.0 represented all pixels within the phantom of the slice of interest being within ± 20% of the global SNR mean value.

### 2.3. In Vivo Data

Acquisition of image data sets using both the RT SUITE and RT AIR coil systems were obtained from healthy volunteers (n = 5) and patients (n = 6) undergoing MR in RT treatment position under an institutional internal review board (IRB)-approved study. For volunteers, immobilization was not used, nor was gadolinium-based contrast administered, and between one and two separate volumetric acquisitions were acquired per coil system. For patients, the standard imaging protocol was obtained in treatment position with the conventional RT SUITE coils, which were then exchanged for the RT AIR coil system for comparative imaging. To maintain a reasonable duration of the imaging session, between one and two volumetric acquisitions were repeated from the original protocol for each patient. All comparison imaging was performed after the administration of intravenous gadolinium-based contrast agents. In all instances, the imaging prescription and scan parameters were identical for the two coil systems. For both patients and volunteers, a single coil configuration for both sets of coils was used. Unlike the phantom data, only global and per slice SNR values were calculated due to the inherent variations in anatomy, pathology, and MR relaxometry across various tissue types, making uniformity measures meaningless. Additionally, because only single acquisitions were acquired for a given pulse sequence and coil combination, the alternate method for estimating the SD of the noise as described in the NEMA standard MS 6-2008 [21] was used, which involved choosing a region of interest (ROI) over a portion of the image that was void of signal and artifacts. Areas where no signal was present (due to the use of gradient nonlinearity distortion correction algorithms) were also avoided. The in-plane dimensions of the noise region of interest (ROI) were fixed at 10 × 10 (X × Y), while the Z dimensions (slice encoding direction) varied between 50 and 100 due to the variable number of imaging slices acquired per subject and pulse sequence.

Individual scan parameters and acquisition conditions for both the patient and volunteer subjects and their respective data are listed in Table A4 of Appendix A.

### 2.4. Statistical Analysis

For both phantom and in vivo data, a one-sample Kolmogorov–Smirnov test (Matlab R2019b, MathWorks, Natick, MA, USA) was performed to test for normality. At the 5% significance level, all data tested rejected the null hypothesis that the data were described by a normal distribution. As a result, a two-sample Kruskal–Wallis test (Matlab R2019b, MathWorks, Natick, MA, USA) was performed to determine if the image quality data for phantom (SNR, uniformity) and in vivo acquisitions (SNR) were from the same or different distributions. This non-parametric analysis of variance (ANOVA) was chosen over a standard ANOVA test that assumes that the data are normally distributed, which was shown to be false based on the Kolmogorov–Smirnov test results. Paired data sets (RT SUITE versus RT AIR) were considered from the same distribution if the returned *p*-value was greater than 0.05 at the 5% significance level. 

## 3. Results

### 3.1. Phantom

The results of the non-parametric Kruskal–Wallis statistical tests are shown in Table A2 of Appendix A for the three image quality tests as a function of coil sets and configurations—namely, the SNR over the entire volume of the phantom (Table A2), the average SNR as a function of image slice (Table A2), and image uniformity (Table A2). Values listed within each cell are the mean ranks (average of ranks for data within a distribution), while shaded cells represent those mean ranks that were not statistically significantly different—that is, from the same distribution. Except for the RTS and AF_AA_PA coil pair comparison, which were determined to be statistically equivalent, all three of the RT AIR coil configurations provided increased volume SNR, per slice SNR, and per slice image uniformity based on their average rank values. Table A2 also demonstrates that global SNR, per slice SNR, and uniformity distributions are affected by the choice of RT AIR coil configuration; however, greater heterogeneity across all three metrics and coil comparisons was identified.

Figure 2a,b show both phantom SNR and uniformity coronal images and plots for the RTS and AR coil type and configuration comparisons. Images represent anterior, mid-volume, and posterior slices and illustrate the impact of coil type (RT SUITE vs. RT AIR), anatomic conformity, and number of coil elements within a given coil on these parameters. Figure 3a–c show the histogram of the global SNR for the two coil types and configurations, as well as the per slice SNR and uniformity metrics. Comparison of both global and per slice SNR plots illustrates an improvement in both metrics when the RT AIR coil is used for imaging. Similar improvements are seen in the uniformity plots; however, these differences are spatially dependent, indicating that this metric is more sensitive to individual differences between coils than both the global and per slice SNR plots.

### 3.2. In Vivo

Table A3 of Appendix A lists the results of the Kruskal–Wallis statistical test for both volunteers and patients as applied to the RT SUITE and RT AIR coil set and configuration comparisons. Distributions were considered statistically nonsignificant (i.e., from the same distribution) if the *p*-value exceeded 0.05 and are identified by the shaded cells. The table includes the results for the global histogram and per slice SNR values paired comparisons for the patient and healthy volunteers enrolled in the study. When comparing SNR within the entire phantom volume, five of the nine and two of the nine distributions for the patients and volunteers were considered from the same distributions, respectively. Comparison of per slice averaged SNR yielded two and one distributions for the patient and healthy subjects that were statistically identical. Differences between distributions are due in part to the different coil configurations used with the RT AIR coil set, which resulted in different numbers of total coil elements per configuration and their anatomic locations relative to the subject. The closest comparison therefore is the RTS and AF coil configurations, which resulted in six paired comparison sets within the patient cohort only. Of these, four of the six and two of the six comparisons for the global SNR and per slice SNR comparisons were statistically the same. 

### 3.3. In Vivo Illustrative Examples

The information contained in Figure 2 and Table A2 and Table A3 provides a quantitative assessment of image quality and allows for an intercomparison of RF coil types and configurations. However, these metrics do not always convey the complex and subtle differences in image quality that are necessary to provide the improved depiction of target volumes and OAR, a prerequisite for precision RT treatment planning. To illustrate these clinically relevant and important differences, several examples are provided in the figures that follow. All comparison images are displayed with identical window and level values.

#### 3.3.1. Margin Delineation

A prerequisite for precision radiation therapy is the ability to decrease treatment margins for both the tumor volume and organs at risk. This requires accurate and precise depiction of these structures throughout the patient’s treatment course (i.e., before, during, and after treatment). Figure 4 illustrates the improved depiction of post-surgical changes in patient subject 4 (P4) following the resection of a pathologically verified glioblastoma multiforme mass within the right frontal lobe, which can be achieved using the AIR RT coil compared to the conventional RT SUITE coil set. The horizontal arrows show the improved depiction of the anterior margin of the operative cavity on post-contrast T_1_-weighted sagittal images obtained using the RT AIR coil (Figure 4b) compared to the RT SUITE coil (Figure 4a). Figure 4b also shows the improved depiction of the small enhancing nodule adjacent to the arrow tip that is not as clearly depicted in Figure 4a due to the increased SNR, despite the increased enhancement in pulsatile flow anterior to the resection cavity and slight nonuniformity across the brain. The fat-suppressed T_2_-weighted images show the improved depiction of the nonenhancing signal changes surrounding the operative cavity due to the higher coil density and improved coil location achieved with the RT AIR coil system (Figure 4d) versus the RT SUITE (Figure 4c). Figure 4d also exhibits a slight loss of image quality in the cerebellum and cervical spine, the source of which is most likely due to swallowing and motion artifacts. However, despite these artifacts and the loss of contrast in the cerebellum, the AIR coil system was able to capture the anatomy within and around the surgical cavity—the anatomic region in question. Statistical analysis of the T_1_-weighted global and per slice SNR showed that they were not statistically significantly different. However, the global SNRs of the T_2_-weighted distributions were with the mean rank of the RT AIR coil being larger than the RT SUITE (269 vs. 231).

#### 3.3.2. Artifact Enhancement

MRI is known to produce a range of imaging artifacts [23,24] that are often more conspicuous in high-resolution, high-SNR MR data. Figure 5 illustrates a subtle ghosting artifact seen in the spinal cord at the level of the cervico-medullary junction on a midline sagittal T_1_-weighted post-contrast-enhanced image from patient subject 5 (P5_C1). There are ghosting and motion artifacts present on both images. Despite this artifact, the remainder of the RT AIR image was deemed superior to the comparative RT SUITE image because of the greater SNR and more homogenous signal within the selected image slice. These localized differences (i.e., as seen within a given imaging slice) also highlight the fact that they are subtle and focal and not reflected in quantitative metrics, as shown in Table A3. The close conformity of the RT AIR coil to the face and mandible results in the improved depiction of the tongue, larynx, and surrounding structures (horizontal arrow) with the RT AIR coil system. Note that the increased signal at the level of the glottis has also resulted in the enhancement of an artifact related to the ghosting signal in the region posterior to the patient (upward arrow). 

#### 3.3.3. Coil Placement

Figure 6 is a comparison midline sagittal slice from a T_2_-weighted fat-saturated volumetric acquisition obtained on a healthy volunteer (V5_C2) and illustrates the impact of coil conformity and selection on regional image quality—in this instance, the prevertebral and paratracheal soft tissues (arrow). The increased signal results from the placement of the anterior array below the chin and the selection of the AF_AA_PA coil configuration. Both global SNR histograms and per slice SNR distributions were statistically significantly different, with the mean rank value of the RT AIR coil being larger than the RT SUITE for both (global SNR: 290 vs. 210, per slice SNR: 170 vs. 118), indicating the superior SNR of the RT AIR coil despite having similar numbers of coil elements in each (28 vs. 30).

#### 3.3.4. SNR and Anatomic Conspicuity

Figure 7 is a comparison of a paramedial sagittal slice from a T_1_-weighted volumetric acquisition of patient 1 (P1_C1) and highlights the effect of increased SNR on lesion conspicuity. The arrow identifies an enhancing small lesion, most likely a small metastatic nodule, more clearly seen on the RT AIR (Figure 7b) image compared to the RT SUITE image (Figure 7a). The subtle increase in lesion conspicuity seen in Figure 7b is not reflected in the global SNR comparison, in which the distributions were not statistically significantly different. While the per slice SNR distributions were statistically different, with the mean RT AIR coil having a higher mean rank than the RT SUITE (175 vs. 145), these differences do not provide the specificity to identify specific slices or regions within a given slice that are statistically significant. 

#### 3.3.5. Image Uniformity and Depiction of Fine Anatomic Detail

Both high SNR and uniform signal intensity are prerequisites for the resolution of fine anatomic detail. Figure 8 illustrates this in the improved depiction of the detail within the operative bed (arrow) of a patient subject (P3_C1) on the representative sagittal T_2_-weighted image. While the RT SUITE (Figure 8a) image shows increased signal intensity anteriorly, the RT AIR (Figure 8b) shows a more uniform signal, allowing a clearer depiction and identification of the margins of the tumor bed and residual, unresected disease. The global SNR and per slice SNR distributions were statistically significantly different between the two coil types, with the mean ranks of the RT SUITE being higher than the RT AIR for both (global SNR: 268 vs. 232, per slice SNR: 206 vs. 98). These seemingly contradictory findings reflect the fact that these statistics do not capture clinically significant, small differences in anatomic regions of high clinical importance, such as the tumor and peritumor volume. Improved image quality can also be seen in the depiction of the tongue (bottom arrow) in the RT AIR coil image. 

The impact of signal uniformity and depiction of fine detail is also illustrated in the sagittal T_2_-weighted images of patient subject 6 (P6_C2), as shown in Figure 9. The arrows identify the parotid duct, which can be clearly identified on the RT AIR image (Figure 9b) compared to the RT SUITE (Figure 9a). Discernment of such anatomic structures that are approaching the in-plane resolution of the image is facilitated by the RT AIR coil’s ability to closely follow the anatomic contours of the patient—in this instance, the mandible. While the global SNR histograms from the two data sets were not statistically significant, the per slice SNR distributions were, with the mean rank of the RT AIR coil being higher than the RT SUITE (170 vs. 151).

## 4. Discussion

The data presented highlight both the challenges and opportunities that exist in MR imaging for RT treatment simulation. RF coil manufacturers are challenged by the need to translate improvements in image quality, which can be easily quantified using standardized metrics such as SNR and signal uniformity in phantoms, into the more complex and often demanding requirements of improving image quality in in vivo MR imaging for RT simulation. This study highlights this in that, in aggregate, new, lightweight RF coils such as the RT AIR coil have been shown to provide quantitative improvements in image quality metrics but that these improvements are more subtle in in vivo clinical imaging scenarios in which the imaging position and body habitus do not allow for a ‘one size fits all’ RF coil design. This is further complicated by the need to meet the conflicting needs of the MR RT simulation process, which requires large anatomic coverage, high resolution, and distortion-free MR data of patients in treatment position while in their immobilization devices. Careful attention is therefore warranted in assessing the performance of new technologies such as the RT AIR coil, both in terms of standardized image quality metrics and their performance under routine clinical imaging conditions. Conversely, the data also identify the potential and opportunity for RT AIR coils and their related technologies to address long-standing challenges of producing high-resolution, high-quality reproducible MR data for RT treatment simulation. Equipment manufacturers therefore need support and encouragement to pursue the development of this and similar technologies to address current limitations and unmet needs of the MR RT imaging community.

While not quantified in this study, the RT AIR coil set and related configurations provide improved ergonomics in terms of coil placement and patient comfort. This is validated by the fact that all patients and volunteers were successfully imaged without failures due to fatigue or discomfort. In addition, the AIR coils were placed by the MR technologists without supervision and therefore not optimized in terms of image quality. This is particularly relevant for the anatomic site studied—the head and neck—which involves the use of tightly fitting thermoplastic immobilization masks. In this context, the AIR coils were extremely forgiving and provided the most flexibility in terms of adaptation to individual body habitus. By contrast, and as seen in Figure 1b,d, two ‘paddle’ coils are placed directly onto the face and mask of the individual, limiting their vision and breathing, and enhancing or inducing claustrophobic sensations. This is further exacerbated by the typical imaging times for these sessions, which can last between 30 and 50 min depending on the site and type of disease. Increased anxiety has the unwanted potential of increasing patient movement, both voluntary and involuntary, resulting in the motion-induced degradation of image quality. The ability to quickly apply surface coils such as the RT AIR set reduces the imaging setup time and the potential for patient motion, while the placement of portals for both the eyes and mouth (Figure 1c) improves patient comfort.

As the AIR coil technology outlined in this work becomes more widely available, research into it and its clinical use are expected to grow. For example, Cogswell et al. [25] described the comparison of a custom-built 16-channel ‘balaclava’ head coil to an 8-channel and 32-channel conventional RF head coil, in which the results demonstrate that an improved SNR can be achieved when these elements closely conform to the patient’s head but that this improvement is dependent upon the number of RF coils; the 16-channel AIR SNR was greater than the 8-channel conventional SNR, while the 16-channel AIR SNR was less than the 32-channel conventional coil. Clinically, Fukui et al. [26] have reported that an improved SNR can be achieved using a commercially available AIR coil when compared to a conventional phased array coil for liver imaging. However, the work failed to disclose the number of AIR coil elements used. In contrast, Bae et al. [27] described the comparison of a 30-channel ‘blanket’ AIR coil to a conventional 16-channel conventional anterior array for ZTE imaging of the lung, which, as expected, indicated increased image quality due in part to the almost doubling in coil elements in the AIR compared to the conventional coil. The unique contribution of this work is the qualitative and quantitative evaluation of a pre-clinical, novel (i.e., AIR) RF coil technology specifically designed to address one of the most challenging anatomic sites for the MR imaging of radiation therapy patients in treatment position, namely the head and neck. Similarities exist between the data presented in this work and those described by Cogswell et al. [25] in that both efforts use the same RF coil technology and apply it to similar anatomies (head versus head and neck). However, this effort addresses a more complex and larger anatomic region (head and neck), with the added complexity of imaging around patient-specific immobilization devices. Further, the applicability of the Cogswell design to MR simulation is limited given that the coil has not been designed for use with immobilization devices, does not include the same anatomic coverage, and is unlikely to be commercialized, thereby limiting its widespread accessibility and use. Finally, Cogswell et al. [25] reported on the evaluation of the prototype AIR coil on the imaging of the brain in healthy volunteers. In contrast, this work reports on the challenges encountered when imaging acutely ill cancer patients, who are less cooperative, are more susceptible to imaging artifacts due to swallowing and breathing, have much more complex anatomy compared to the brain, and are more prone to degraded image quality due to anatomy-induced susceptibility differences that are exacerbated at 3.0 T compared to 1.5 T.

The data presented herein successfully demonstrate the feasibility of using the RT AIR coil system in a clinical setting. However, there are several limitations to further generalizing the results of this study. First, for a given coil set and configuration comparison, no two had the same number of RF coil elements, nor were individual elements positioned over the same exact anatomy. This means that differences in signal intensity and uniformity are inherent and may therefore bias the results both in terms of phantom and in vivo testing. For example, images reconstructed with the RT SUITE configuration involve combining signals from 30 independent coil elements. The RT AIR coil configuration AF_AA_PA includes 28 coil elements but only uses signals from the bottom four elements from the posterior RT open array and two from the spinal array embedded in the MR table. This results in reduced signal posteriorly when using the AF_AA_PA compared to the RT SUITE configuration, impacting both the SNR and uniformity within the brain. Similarly, the RT AIR configuration provides an additional signal inferior to the location of the RT open array, which translates into an increased posterior signal at the level of the C7-T1 vertebral junction. The different form factors and placement of the individual coil components also impact image quality and performance, as well as the application of image intensity corrections, which were applied to the in vivo data, further affecting the image quality and potentially biasing in vivo coil comparison data. These differences do, however, highlight the clinical reality of MR RT treatment simulation of having limited dedicated RF coils and coil configurations. Within this context, the comparisons represent a ‘real-world’ scenario and therefore represent findings that are translatable into routine clinical imaging.

Second, the study did not assess or quantify the effects of differences between coil types and configurations for the in vivo patient and volunteer subject data on image quality, or dosimetric differences resulting from differences in the contouring of target volumes and adjacent OAR. Rather, the study reported selected examples of observed differences between the various combinations, with the aim of identifying and highlighting specific anatomical features that are either enhanced or degraded depending upon the RF coil used under typical imaging conditions. Statistical analysis of in vivo SNR distributions provides insight into the performance of respective coil sets and configurations and gives guidance as to which are best suited for individual anatomical sites and disease processes, but does not capture the complete performance of a given coil set. In addition, patient data were obtained after the administration of gadolinium contrast agents, which, by their nature, have a time-dependent effect on image contrast. While all patient subjects were imaged immediately and sequentially with the various coil combinations, the average delay between the repetition of identical sequences was, on average, 13 min. The effect on T_2_-weighted image quality is expected to be minimal but could be impactful on T_1_-weighted data. However, most patient subject comparisons involved T_2_-weighted image sets (T_2_ = 6 comparisons, T_1_ = 3 comparisons). To fully evaluate the clinical performance of these coil types, one requires blinded comparisons of image quality involving multiple observers, scoring, and statistical analysis, as well as the assessment of the dosimetric impact resulting from differences, if any, in the contouring of target volumes and OAR delineation across the various image sets. While the quantification of differences in target delineation and dosimetric effects are the ultimate metrics for evaluating these coils in the setting of MR simulation for RT treatment planning, this process is beyond the scope of the current work but is an endeavor of future research efforts. 

Third, the RT AIR coil evaluated in this study requires further hardware and software optimization to realize fully the advantages of this technology. The current RT AIR coil represents a hybrid RF coil that combines conventional coil elements embedded into the posterior open array and posterior spine array embedded into the MR table. While it has been demonstrated that both AIR coil and conventional RF coil loops of approximately equal dimensions provide similar imaging characteristics (SNR, depth of penetration), AIR coil elements exhibit significantly less mutual inductance [19], resulting in lower geometry factors (i.e., g-factors), which affects the SNR when parallel imaging is employed (higher g-factors result in lower SNR), thereby allowing for higher coil densities and a resultant increase in image quality. As a result, the current RT AIR coil configuration does not represent the highest attainable image quality that could be achieved by replacing conventional RF coils with AIR coils. Replacement of conventional RF coils with AIR coil elements is likely to further improve the overall performance of the RT AIR coil system.

## 5. Conclusions

Ongoing advances in precision radiotherapy are generating an increasing demand for higher-quality imaging data sets to ensure the more accurate and precise delineation and characterization of both tumor volume and signal characteristics. MR imaging for treatment simulation and planning has great potential to meet this need, and the further implementation of recently introduced AIR coil technology can further advance the quality of dedicated RT planning MR with patients in treatment position. 

This study indicates that a new, highly flexible, and lightweight RF coil system improves image quality in both phantom and human subjects undergoing MR imaging for RT treatment simulation and planning of the head and neck. While not quantitatively assessed, initial experience indicates that the RT AIR coil system provides improvements in in vivo imaging setup and patient compliance compared to the conventional RF coil system used in routine clinical use within our practice. 

## Figures and Tables

**Figure 1 jcm-11-05984-f001:**
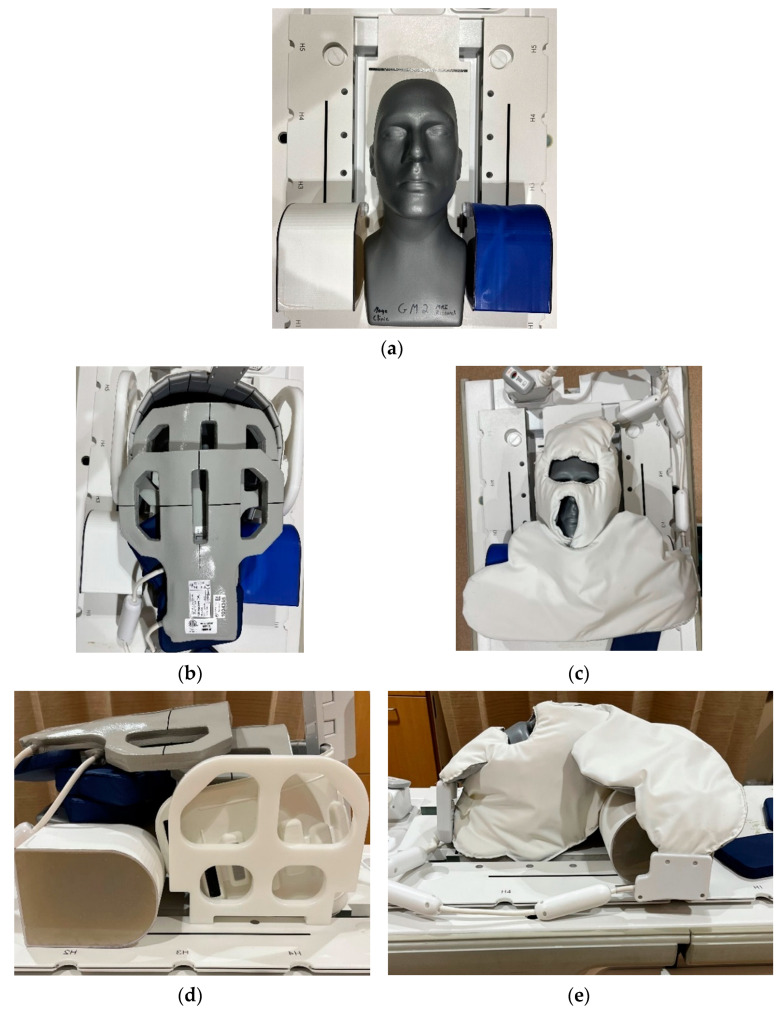
Phantom evaluation of the two RF coil systems. (**a**) Humanoid (gray) and shoulder mimicking phantoms. (**b**) Anterior and (**d**) lateral views of the placement of the RT SUITE RF coils. (**c**) Anterior and (**e**) lateral views of the placement of the RT AIR coil set. The RT AIR coil configuration includes the 15-element array designed to cover the face and head and seven-element anterior array covering the shoulders and brachial plexus. Both coil sets include the use of the posterior eight-element open array located below the CIVCO immobilization board.

**Figure 2 jcm-11-05984-f002:**
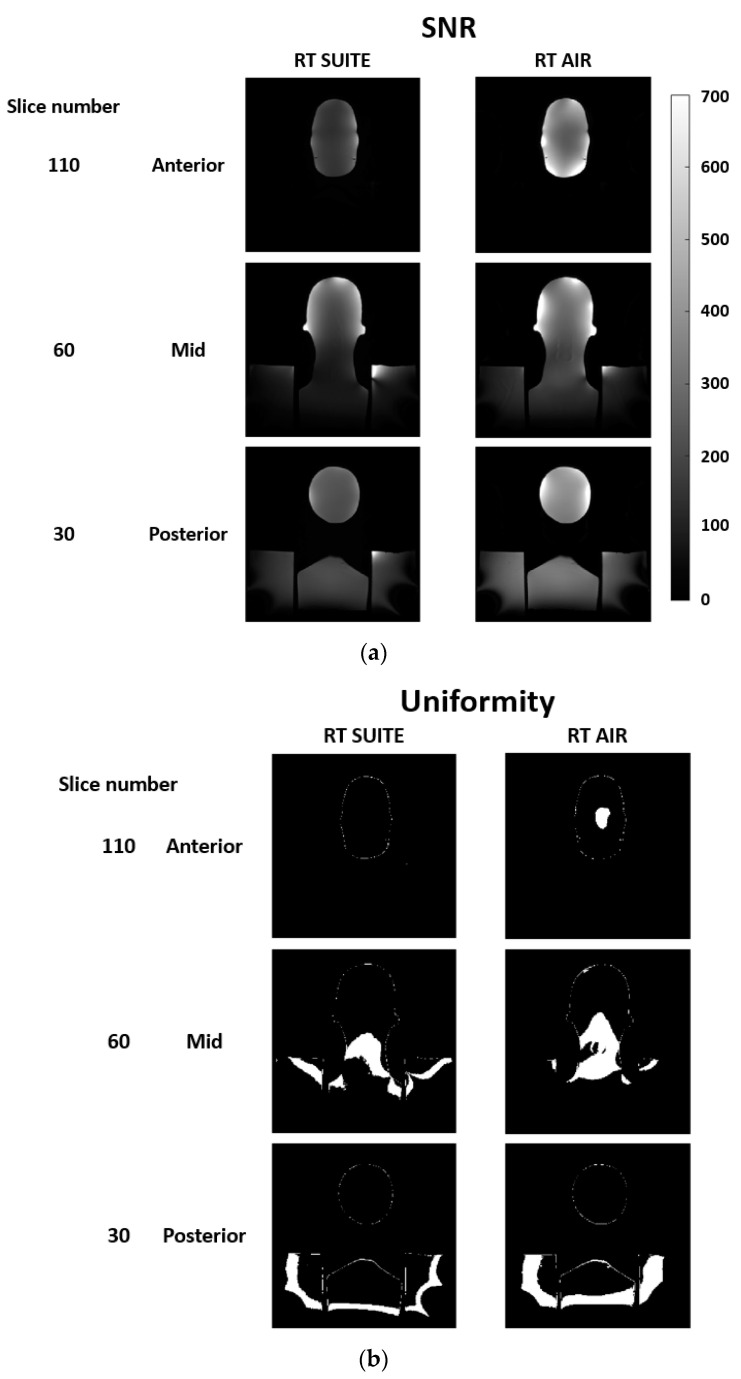
Representative coronal slices through the anterior, mid, and posterior of the phantom used to evaluate image quality. (**a**) Columns one and two are signal to noise ratio (SNR) images at the three slice locations for the RT SUITE (column 1) and RT AIR (column 2) coils. (**b**) Uniformity images at the three slice locations for the RT SUITE (column 1) and RT AIR (column 2). SNR values within ±20% of mean SNR within the volume are set to 1.0, while those outside of this range are set to zero. The coil configuration AF was used for the RT AIR coil acquisition. SNR figures are displayed with the same window and level values (600, 300).

**Figure 3 jcm-11-05984-f003:**
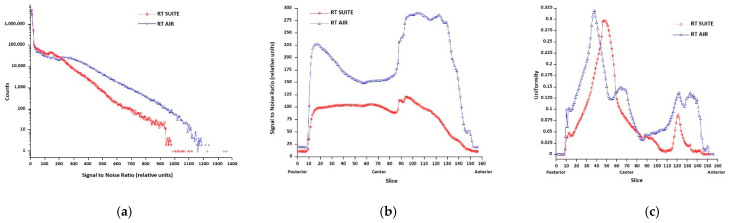
Phantom SNR and uniformity quality metric plots for the RT SUITE and RT AIR (configuration = AF) coil systems. (**a**) SNR histogram, which includes SNR values within the entire phantom distributed across 250 gray level bins ranging from the minimum to maximum SNR value. (**b**) Per slice average SNR and (**c**) uniformity for the two coil sets are also shown. Slice numbers range from posterior (slice number 1) to anterior (slice number 156), with the center of the phantom volume being slice number 70.

**Figure 4 jcm-11-05984-f004:**
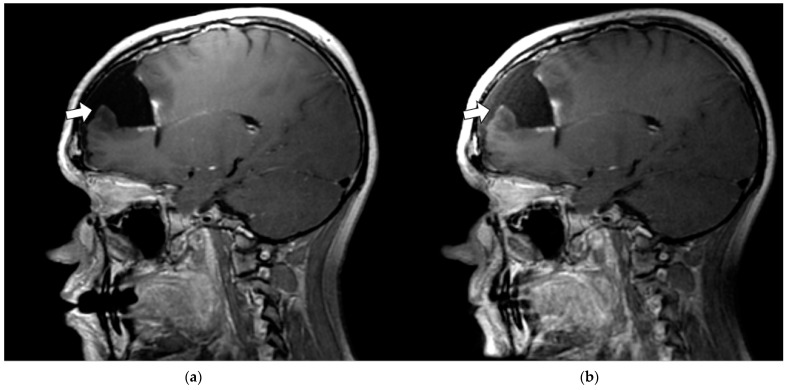
Sagittal slice from both T_1_- and T_2_-weighted volumetric imaging of patient subject 4 (P4_C1 and P4_C2). (**a**) T_1_-weighted RT SUITE. (**b**) T_1_-weighted RT AIR. (**c**) T_2_-weighted RT SUITE. (**d**) T_2_-weighted RT AIR. The anterior margin of the resection cavity (horizontal arrows) with small enhancing lesion is better appreciated with the AIR coil system (**b**) because of increased SNR and improved signal uniformity. The nonenhancing hyperintense T_2_ signal surrounding the operative bed (vertical arrows) is better delineated on the RT AIR image.

**Figure 5 jcm-11-05984-f005:**
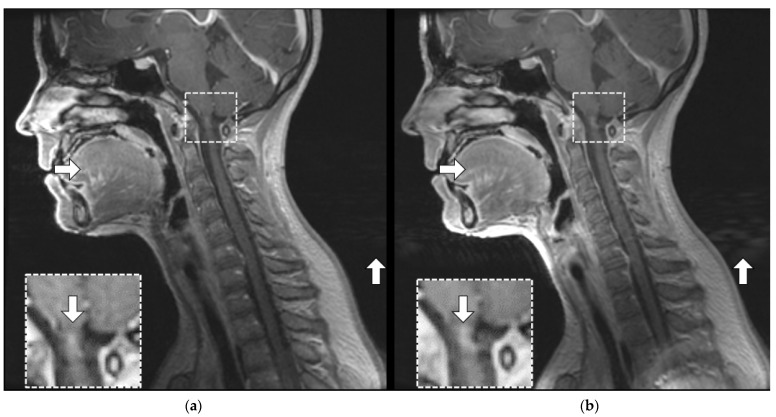
Post-gadolinium contrast sagittal T_1_-weighted slices from patient subject 5 (P5_C1). (**a**) T_1_-weighted RT SUITE. (**b**) T_1_-weighted RT AIR. The dashed rectangle identifies tissue mimicking signal artifact (down arrow) at the base of skull within the spinal cord, which is seen more prominently in the RT AIR image (**b**) compared to the RT SUITE (**a**). The horizontal arrow shows the improved depiction of the tongue and related structures in (**b**) compared to (**a**), while the vertical (upward) arrow identifies motion-induced artifacts due to motion of the glottis following swallowing.

**Figure 6 jcm-11-05984-f006:**
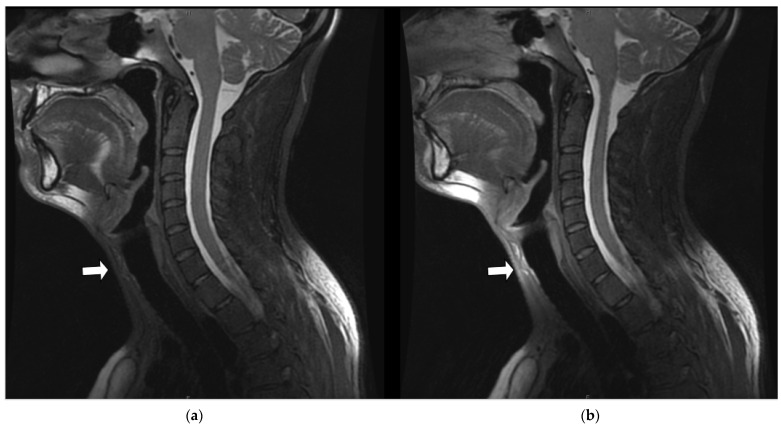
Midline sagittal slice from T_2_-weighted fat-saturated volumetric acquisition from volunteer 5 (V5_C2). (**a**) T_2_-weighted RT SUITE. (**b**) T_2_-weighted RT AIR. Note that the horizontal arrow identifies the increased signal in the anterior neck and glottis achieved with the RT AIR coil (**b**) compared to the RT SUITE (**a**) (horizontal arrows). The increased signal available with the RT AIR coil in (**b**) is a result of the activation of the 7-element anterior array, as well as the close conformity of the 15-element face array of the RT AIR coil (coil configuration = AF_AA_PA).

**Figure 7 jcm-11-05984-f007:**
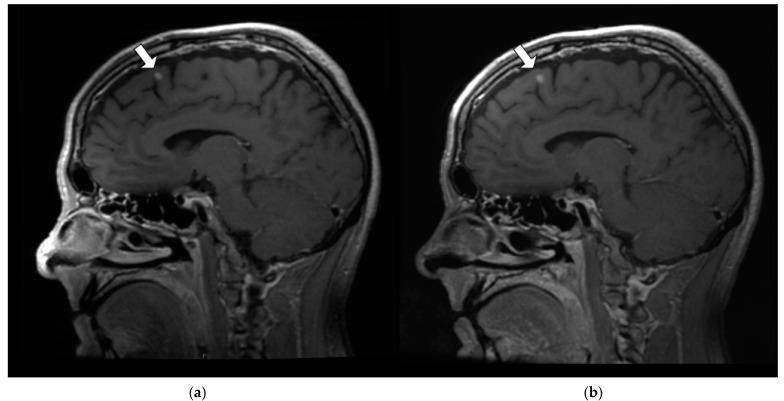
Paramedian sagittal slice from T_1_-weighted fat-saturated volumetric acquisition from patient 1 (P1_C1). (**a**) T_1_-weighted RT SUITE. (**b**) T_1_-weighted RT AIR. Note the improved depiction of a suspected metastatic lesion (arrows) on the RT AIR coil image (**b**) compared to the RT SUITE image (**a**). There was also comparatively improved SNR within the palate, tongue, and prevertebral regions on the RT AIR coil image.

**Figure 8 jcm-11-05984-f008:**
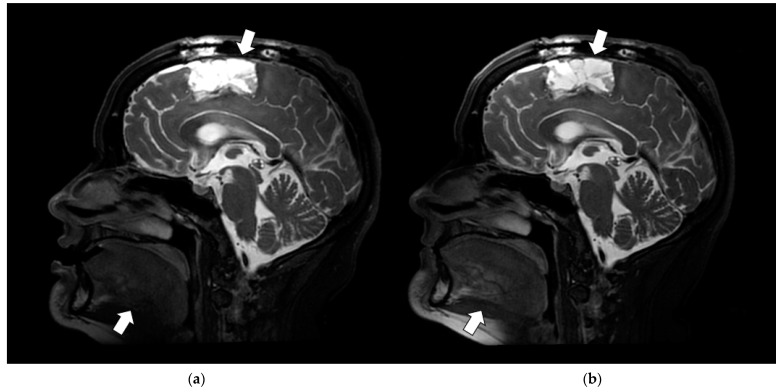
Sagittal slice from T_2_-weighted fat-saturated volumetric acquisition from patient 3 (P3_C1). (**a**) T_2_-weighted RT SUITE. (**b**) T_2_-weighted RT AIR. Note the improved depiction of fine detail within the posterior right frontal operative cavity (top arrow) with the RT AIR coil (**b**) compared to the RT SUITE coil set (**a**) when displayed with the same image window and level values. The RT AIR coil also shows improved depiction of the tongue and soft pallet (bottom arrow) due to the use of the 7-element anterior array.

**Figure 9 jcm-11-05984-f009:**
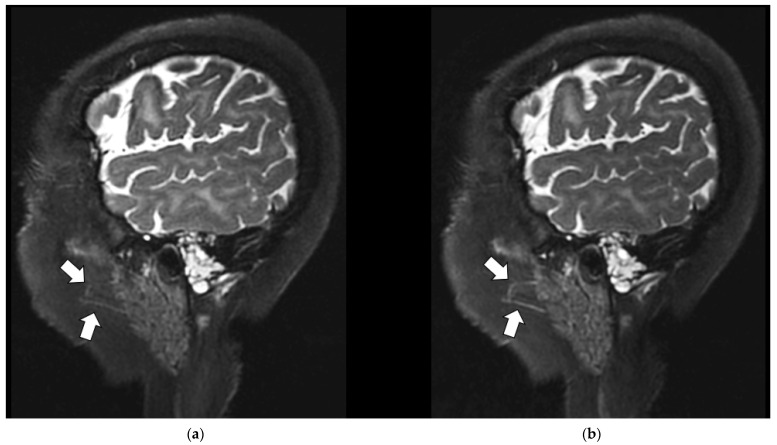
Sagittal slice from T_2_-weighted fat-saturated volumetric acquisition from patient 6 (P6_C2). (**a**) T_2_-weighted RT SUITE. (**b**) T_2_-weighted RT AIR. Improved depiction of the parotid gland architecture and parotid duct (arrows) is achieved with the RT AIR coil (**b**) relative to the RT SUITE coil (**a**).

## Data Availability

The data in this study are not publicly available.

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
