# Peer review of "Evaluation of a New, Highly Flexible Radiofrequency Coil for MR Simulation of Patients Undergoing External Beam Radiation Therapy"

_jcm, 2022, doi:10.3390/jcm11205984_

Round 1

Reviewer 1 Report

The paper presents a novel technology which was validated both on models and in human, thus there is no need to improve the merit of the paper, but rather the structure of the future article. 

1. I would move the tables from the main text to the supplementary files and present only the most essential findings in form of a graph. 

2. I suggest including the table which would compare Authors' coil to standard used in clinics. 

3. There is no information about the consent of the ethics committee for the study. Please add such information. 

4. The whole paper is not formated in MDPI standards.

All in all, the paper is very interesting, but the technical side of the manuscript should be revised.

Author Response

Reviewer 1

Authors response to reviewer comments

The authors would like to thank the reviewer for their insightful review. Listed below is a response to each comment that we believe successfully and succinctly addresses each comment / concern. In addition, the response to all 3 reviewer’s comments in aggregate have further strengthened the work to the point that it is acceptable for publication.

For the reviewer’s convenience the review has been reproduced below with the response to each comment by the authors shown in red.

Reviewer 1 comments

Open Review

English language and style

( ) Extensive editing of English language and style required
( ) Moderate English changes required
(x) English language and style are fine/minor spell check required
( ) I don't feel qualified to judge about the English language and style

Yes

Can be improved

Must be improved

Not applicable

Does the introduction provide sufficient background and include all relevant references?

(x)

( )

( )

( )

Are all the cited references relevant to the research?

(x)

( )

( )

( )

Is the research design appropriate?

(x)

( )

( )

( )

Are the methods adequately described?

(x)

( )

( )

( )

Are the results clearly presented?

( )

(x)

( )

( )

Are the conclusions supported by the results?

(x)

( )

( )

( )

Comments and Suggestions for Authors

  1. The paper presents a novel technology which was validated both on models and in human, thus there is no need to improve the merit of the paper, but rather the structure of the future article. 

The authors have attempted to reformat the paper to improve readability in line with the reviewer’s recommendations.

  1. I would move the tables from the main text to the supplementary files and present only the most essential findings in form of a graph. 

The tables have been moved to Appendix A of the document. The authors are unclear as to how to present the results of the data listed in the tables in graphical form and ask that the reviewer provide more insight into how this could be achieved.

  1. I suggest including the table which would compare Authors' coil to standard used in clinics. 

The text has been modified to describe the vendor’s proposed coil positioning.

Page 4, line 103 now reads:

This does not represent the placement recommended by the manufacturer which consists of using a custom coil positioning device in which each paddle array is placed along both sides of the subject’s head and the rectangular array suspended above the patient’s clavicles with the coil’s longest dimension left-to-right. Instead, the positioning used in this work is

  1. There is no information about the consent of the ethics committee for the study. Please add such information. 

IRB statement is included in the section entitled Institutional Review Board Statement (Page 22, line 555). The exact text is:

The study was conducted according to the guidelines of the Declaration of Helsinki and approved by the Institutional Review Board of Mayo Clinic, Rochester, MN, IRB # 16-010554, initial approval granted January 30, 2017.

In addition, the Methods section includes the following statement (page 7, line 191):

Acquisition of image data sets using both the RT SUITE and RT AIR coil systems were acquired from healthy volunteers (n=5) and patients (n=6) undergoing MR in RT treatment position under an institutional internal review board (IRB) approved study.

  1. The whole paper is not formatted in MDPI standards.

The authors are not clear on what the reviewer is referring to as the manuscript was prepared using the journal’s template and includes all sections in the same order and format (Introduction, Materials and Methods, Results, Discussion, Conclusions). The authors have re-reviewed the work and believe that the manuscript is in compliance with the standards identified by the journal. The authors would like to request that the reviewer provide additional details regarding formatting inconsistencies.

Reviewer 2 Report

The manuscript “Evaluation of a new, highly flexible radiofrequency coil for MR simulation of patients 2 undergoing external beam radiation therapy” presents a comparison among two commercially available multi-coil setups for head and neck imaging. The manuscript’s purpose is to test the two coils in a specific clinical application: imaging for RT planning.

If a manuscript must evaluate the clinical performance of a device it requires “blinded comparisons of image quality involving multiple observers, scoring and statistical analysis” as the authors recognize in the Discussion. If instead a comparison is performed for an audience involved in the technical developments of MRI hardware it is usually more based on different quantitative evaluations of the coils’ performances (like SNR, B1 homogeneity), often performed with uniform phantoms to avoid patient’s variability. Present manuscript is not directed to evaluate the clinical performances but contains a long list of exemplar cases where the authors focus on subtle differences in the images which are qualitative and without any evident clinical interest.

For this reason, I invite the author to better focus on their target audience in the revised version.

In the following a detailed list of points that should be revised:

Introduction

-       Line 80: Please add references for the AIR coil technology

Methods And Materials

-       Line 97: ”Figure 1a” should instead refer to Figure 1a, b, d

-       Line 102: “Figure 1b” should instead refer to Figure 1c, e

-       Table 1: Please add a vertical line to separate the RT SUITE and RT AIR sectors of the table to help the reader.

-       Line 180: “This value was then divided into the first of the two imaging volumes to create a 3D SNR map.” Sorry, I couldn’t understand the meaning of this sentence.

Results

-       Line 224: Figure 2(c) is not present.

-       Line 287: The authors declare they acquired 5 patients, but Table 3 contains result for Patient P6 as well.

-       Line 288: I do not understand the meaning of “C1 – C3 = comparison 1 – 3”.

-       Line 331: “… the remainder of the RT AIR image was deemed superior to the comparative RT SUITE image because of greater SNR and more homogenous signal” This sentence seems not to agree with results of Table 3 where P5-C1 line shows no significative difference among coils for Global SNR parameter and reduced performance of RT AIR for per Slice SNR parameter.

-       Line 370: “The subtle increase in lesion conspicuity seen in (b) is not reflected in the global SNR comparison in which the distributions were not statistically significantly different.” Here the authors recognize that the global parameters they evaluated could not be relevant in the assessment of a clinical outcome (the lesion detection in this example). If so, which relevance the in-vivo examples presented here have? How could we exclude that it is possible to bring as many examples pointing right in the opposite direction, i.e. towards a better performance of RT SUITE ?

-       Line 386: “While the RT SUITE (a) image shows increased signal intensity, the RT AIR (b) shows more uniform signal allowing clearer depiction and identification of the margins” Here the issue could come from a non-optimal windowing of the RT SUITE image levels. This statement sounds too subjective.

-       Line 390: “These seemingly contradictory findings reflect the fact that these statistics do not capture clinically significant small differences in anatomic regions of high clinical importance”  
The results in Table 2 and Table 3 show a mild support towards the AIR technology. If we focus on statistically significant differences (non-shaded cells) the scores are higher for RTS and AIR respectively accordingly to the following scheme (RTS-AIR):
   Table 2, 1-2 for Volume SNR, 2-3 for slice SNR, 0-5 for slice Uniformity;
   Table 3, 3-8 for Global SNR, 6-9 for Slice SNR.
If, on top of that, there is no well-established and validated evidence of superior clinical outcome how can we conclude in favor of a specific technology?

-       Line 406: “Note the improved depiction of fine detail within the posterior right frontal operative” please see the comment to Line 386.

-       Line 407: “The RT AIR coil also shows improved depiction of the tongue and soft pallet”
Ok, but has it a general interest for the clinical application?

-       Line 413: “Improved depiction of the parotid gland architecture and parotid duct (arrows)” see the previous comment.

Discussion

-       Line 442: “The ability to quickly apply surface coils such as the RT AIR set reduce imaging set up time and the potential for patient motion”
I understand that AIR can reduce set up time but please provide more arguments why it should reduce motion artefacts.

-       Line 467: “Rather, the study reported selected examples of observed differences between the various combinations with the aim of identifying and highlighting specific anatomical features that are either enhanced or degraded”
Ok, but are these features of general clinical interest for head-neck RT planning?

-       Line 475: “the average delay between repetition of identical sequences was on average 13 minutes”
I understand 13 minutes is the coil set-up time but the time difference among acquisition consists of the time duration of the acquisition itself (30 to 50 minutes as from Line 440) plus the set up time of the second coil, for a total of about one hour. This is the time the authors should consider in their discussion of delay among acquisitions.

Author Response

Reviewer 2

Authors response to reviewer comments

The authors would like to thank the reviewer for their insightful review. Listed below is a response to each comment that we believe successfully and succinctly addresses each comment / concern. In addition, the response to all 3 reviewer’s comments in aggregate have further strengthened the work to the point that it is acceptable for publication.

For the reviewer’s convenience the review has been reproduced below with the response to each comment by the authors shown in red.

Reviewer 2 Comments

Open Review

English language and style

( ) Extensive editing of English language and style required
( ) Moderate English changes required
(x) English language and style are fine/minor spell check required
( ) I don't feel qualified to judge about the English language and style

Yes

Can be improved

Must be improved

Not applicable

Does the introduction provide sufficient background and include all relevant references?

( )

( )

(x)

( )

Are all the cited references relevant to the research?

(x)

( )

( )

( )

Is the research design appropriate?

( )

(x)

( )

( )

Are the methods adequately described?

(x)

( )

( )

( )

Are the results clearly presented?

( )

(x)

( )

( )

Are the conclusions supported by the results?

( )

(x)

( )

( )

Comments and Suggestions for Authors

The manuscript “Evaluation of a new, highly flexible radiofrequency coil for MR simulation of patients 2 undergoing external beam radiation therapy” presents a comparison among two commercially available multi-coil setups for head and neck imaging. The manuscript’s purpose is to test the two coils in a specific clinical application: imaging for RT planning.

If a manuscript must evaluate the clinical performance of a device it requires “blinded comparisons of image quality involving multiple observers, scoring and statistical analysis” as the authors recognize in the Discussion. If instead a comparison is performed for an audience involved in the technical developments of MRI hardware it is usually more based on different quantitative evaluations of the coils’ performances (like SNR, B1 homogeneity), often performed with uniform phantoms to avoid patient’s variability. Present manuscript is not directed to evaluate the clinical performances but contains a long list of exemplar cases where the authors focus on subtle differences in the images which are qualitative and without any evident clinical interest.

For this reason, I invite the author to better focus on their target audience in the revised version.

The authors agree with the reviewer’s perspective regarding clinical performance. This is why the discussion states explicitly (see Page 21, Line 511):

To fully evaluate the clinical performance of these coil types requires blinded comparisons of image quality involving multiple observers, scoring and statistical analysis as well as the assessment of the dosimetric impact resulting from differences, if any, in the contouring of target volumes and OAR delineation across the various image sets.

The authors do however state that the purpose of the work is to evaluate the performance the new coil in the first sentence of the abstract (Page 1, Line 15). As such, the authors believe that the manuscript clearly describes the purpose of the work without making claims beyond this (i.e., clinical performance). Given that the manuscript is in response to a special issue of the journal related to MR imaging in Radiation Therapy, the authors believe that the work precisely targets this audience.

In the following a detailed list of points that should be revised:

Introduction

-       Line 80: Please add references for the AIR coil technology

 Relevant AIR coil references added. See Page 3, line 84

Methods And Materials

-       Line 97: ”Figure 1a” should instead refer to Figure 1a, b, d

 Text modified to reflect specific coils. See Page 4, line 102

-       Line 102: “Figure 1b” should instead refer to Figure 1c, e

 Text modified to reflect specific coils. See Page 4, line 109

-       Table 1: Please add a vertical line to separate the RT SUITE and RT AIR sectors of the table to help the reader.

 The authors agree that vertical lines would help in Table 1. However, the journal does not allow for this formatting detail.

-       Line 180: “This value was then divided into the first of the two imaging volumes to create a 3D SNR map.” Sorry, I couldn’t understand the meaning of this sentence.

 Text modified to provide further clarification on the SNR calculation method and is as follows (See Page 6, line 170…):

An SNR volume was calculated by first measuring the standard deviation (SD) of the difference between the sequential T1-weighted 3D CUBE acquisition described previously. The SD value was calculated over a region of interest within the difference (i.e. subtracted) volume equal to 40 x 170 x 10 (X x Y x Z) pixels centered at the reference position (center of phantom and anatomic landmark) in accordance with the recommendations provided by the NEMA recommended SNR measurement protocols [21,22]. 

Results

-       Line 224: Figure 2(c) is not present.

 Text corrected and now refers to Tables A2(a), (b) and (c). See page 8, lines 229 – 230.

-       Line 287: The authors declare they acquired 5 patients, but Table 3 contains result for Patient P6 as well.

 Text corrected and now refers patients 1 – 6. See page 23, line 605.

-       Line 288: I do not understand the meaning of “C1 – C3 = comparison 1 – 3”.

Text modified and now reads (See Page 25, Line 599):

C1 – C3 = comparison 1 – 3 where the C refers to the comparison of two imaging volumes of a given contrast type and the numerical value the number of contrasts/sequences evaluated

-       Line 331: “… the remainder of the RT AIR image was deemed superior to the comparative RT SUITE image because of greater SNR and more homogenous signal” This sentence seems not to agree with results of Table 3 where P5-C1 line shows no significative difference among coils for Global SNR parameter and reduced performance of RT AIR for per Slice SNR parameter.

The authors have tried to reflect the fact that many of the improvements in image quality are regional and focal. To that end, text modified and now reads (See Page 14, Line 328):

There are ghosting and motion artifacts present on both images. Despite this artifact, the remainder of the RT AIR image was deemed superior to the comparative RT SUITE image because of greater SNR and more homogenous signal within the selected image slice. These localized differences (i.e., as seen within a given imaging slice) also highlight the fact that they are subtle and focal and not reflected in quantitative metrics as shown in Table A3.

-       Line 370: “The subtle increase in lesion conspicuity seen in (b) is not reflected in the global SNR comparison in which the distributions were not statistically significantly different.” Here the authors recognize that the global parameters they evaluated could not be relevant in the assessment of a clinical outcome (the lesion detection in this example). If so, which relevance the in-vivo examples presented here have? How could we exclude that it is possible to bring as many examples pointing right in the opposite direction, i.e. towards a better performance of RT SUITE ?

As noted in the previous comment, many of the improvements in image quality are subtle and require expert (i.e. Radiologist) interpretation to highlight them which is the reason for providing them in the first place. However, it is known that improvements in image quality, specifically quantitative measures such as image uniformity and SNR are correlated with an increase in their respective values. In fact, this is a major motivating factor for imaging patients at higher field strengths (3.0T vs. 1.5T for example). The purpose of the phantom analysis is to provide an unbiased comparison of RF coil performance. The in vivo examples give examples of improvements in image quality as noted by highly qualified experts (i.e. neuroradiologists) and to provide quantitative analysis (SNR) for comparison.

-       Line 386: “While the RT SUITE (a) image shows increased signal intensity, the RT AIR (b) shows more uniform signal allowing clearer depiction and identification of the margins” Here the issue could come from a non-optimal windowing of the RT SUITE image levels. This statement sounds too subjective.

The legend of Figure 8 has been modified to read as follows (see Page 16, line 407):

Note the improved depiction of fine detail within the posterior right frontal operative cavity (top arrow) with the RT AIR coil (b) compared to the RT SUITE coil set (a) when displayed with the same image window and level values.

-       Line 390: “These seemingly contradictory findings reflect the fact that these statistics do not capture clinically significant small differences in anatomic regions of high clinical importance”  
The results in Table 2 and Table 3 show a mild support towards the AIR technology. If we focus on statistically significant differences (non-shaded cells) the scores are higher for RTS and AIR respectively accordingly to the following scheme (RTS-AIR):
   Table 2, 1-2 for Volume SNR, 2-3 for slice SNR, 0-5 for slice Uniformity;

   Table 3, 3-8 for Global SNR, 6-9 for Slice SNR.
If, on top of that, there is no well-established and validated evidence of superior clinical outcome how can we conclude in favor of a specific technology?

Tables A2 and A3 report on the quantitative (i.e., statistical) performance of not only the two coil types (AIR vs. RT SUITE) but also on the various coil configurations provided by the manufacturer for the AIR coil. As noted in the Methods and tabulated in Table A1 and discussed in the Discussion section (Page 20, line 479 onwards) coil performance is dependent not only on the type of coil but also the number of elements within the coil (i.e., configuration). Taken in aggregate, these data demonstrate that improved image quality can be achieved with the AIR coils when compared to the RT SUITE set but that overall image quality is dependent upon the specific anatomic site and the chosen coil configuration selected at the time of imaging (see Page 1, line 31).

The purpose of the work, as described in the Introduction was to test the hypothesis that ‘...a novel prototype coil system that uses a recently introduced RF coil technology known as adaptive image receive (AIR) coil system would be ideally suited for MR in treatment position of head and neck RT simulations compared to traditional RF coil systems.’. Importantly, the authors do not comment on how this new technology will affect clinical outcomes and note that further study is required (see Page 21, lines 511 – 517). However, it does represent a first step towards that goal by providing quantitative assessment of the AIR coil system compared to an existing system. The authors also point to the development of the MR-linac which has been developed prior to validation of clinical outcome on the assumption that improved image contrast and quality (i.e., MR vs CT) will result in improved outcomes due to more precise targeting of the tumor volume and adjacent organs at risk.

-       Line 406: “Note the improved depiction of fine detail within the posterior right frontal operative” please see the comment to Line 386.

The authors have addressed the concern in the comment noted earlier.

-       Line 407: “The RT AIR coil also shows improved depiction of the tongue and soft pallet”
Ok, but has it a general interest for the clinical application?

This depends on what the location of the disease. For example, a mass within the mouth and/or tongue would be more clearly visualized with the AIR coil.

-       Line 413: “Improved depiction of the parotid gland architecture and parotid duct (arrows)” see the previous comment.

As above, this depends on what the location of the disease.

Discussion

-       Line 442: “The ability to quickly apply surface coils such as the RT AIR set reduce imaging set up time and the potential for patient motion”
I understand that AIR can reduce set up time but please provide more arguments why it should reduce motion artefacts.

This was described in the following sentence (Page 19, line 443):

By contrast and as seen in Figures 1(b) and (d), two ‘paddle’ coils are placed directly onto the face and mask of the individual limiting their vision, breathing, and enhancing or inducing claustrophobic sensations.

-       Line 467: “Rather, the study reported selected examples of observed differences between the various combinations with the aim of identifying and highlighting specific anatomical features that are either enhanced or degraded”
Ok, but are these features of general clinical interest for head-neck RT planning?

The authors are of the opinion that the answer to this question is yes. This is based on the reality that head and neck cancers are not confined to a specific anatomic location. As a result, high quality images must be generated in which all anatomy is clearly visualized.

-       Line 475: “the average delay between repetition of identical sequences was on average 13 minutes”
I understand 13 minutes is the coil set-up time but the time difference among acquisition consists of the time duration of the acquisition itself (30 to 50 minutes as from Line 440) plus the set up time of the second coil, for a total of about one hour. This is the time the authors should consider in their discussion of delay among acquisitions.

The authors are referring to the imaging times encountered in our clinical practice that do not involve the use of two separate RF coils. To that end, the sentence has been modified to read (see Page 19, Line 445):

This is further exacerbated by typical imaging times for these sessions that can last between 30 and 50 minutes depending on the site and type of disease.

Reviewer 3 Report

In the presented manuscript the authors compare different coil systems for imaging the head at a field strength of 3T with the specific requirement that the posture of the patient be adopted for RT treatments. The comparison is carried out using phantom and patient measurements and evaluated using SNR and image uniformity as the major figure of merit.

General comments

1.     The original ISMRM abstract detailing the coil configuration is cited. However, since then a number of manuscripts have evaluated the performance of the AIR coils for different body regions including head imaging, e.g. DOI: 10.2214/AJR.20.22812 to mention just one. This also compares SNR in the head vs. commercially available head coils and the authors need to clearly point out the novelty of their investigation.

2.     Given a suitably well-designed coil we would expect better SNR ratios if the receiver arrays are “conformally” arranged with the patient’s skin (of have less distance to the patient than possible with rigid coils). So, this result can be expected. Given that the comparison with the commercial coil uses some flexible arrays as well, the results are somewhat confounding. How much SNR is gained from just placing the AIR coils closer to the patient and how much of the SNR gain is due to other factors like number of elements.

3.     One more interest that is of practical interest (at least for everybody doing MR examinations) is the question of patient preparation. While I think that we have a mature workflow using “conventional” coils (rigid and semi-rigid) it would be interesting to hear about the experience of the authors about using the “cloth-like” AIR coils on patients.

Specific comments

1.     The authors frequently talk about “patients undergoing MR simulation”. My apologies, but I am not familiar with the term nor can I guess its meaning.

2.     Introduction, lines 59-60: I agree that commercial coils are not as flexible as AIR coils but they can be suitably arranged around the patient’s head – albeit maybe with a lower number of receiver numbers. Maybe this could be rephrased for clarity.

3.     Methods and Materials, lines 109-110: Yes, one would need to modify the coil files to adopt the system for an alternative coil configuration (be aware that be doing so the MR system uses FDA approval). However, the interesting question here would be how the signal from the different coil systems and individual elements is combined for image calculation.

4.     Methods and Materials, lines 130-135: Could the authors provide details about the liquid used to fill the phantom and its composition?

5.     Materials and Methods, lines 146-147 and 167-178: Actually, the referenced NEMA standard does provide multiple methods to calculate SNR from MR images. Form the description provided it is not clear which one was used. I can only guess that “two sequential” measurements imply a measurement without the phantom to get the SD of the noise pixels and a subsequent second measurement with the phantom for the mean signal intensity? How are the 3D maps of SNR obtained?

6.     Results: Table 2 seems to indicate some performance benefits for the AIR coil configurations. However, Fig. 3b shows better per slice SNR performance for the RT suite configuration. Please explain.

7.     Results: Figure 2c (assuming white means the region of uniformity) shows higher uniformity for the AIR coil configurations. This does not seem to be backed up by Figure 3(c) where (except for a small range of slices) the RT suite configurations performs better. Please explain.

8.     Results Figure 2(a); Please add scales to the SNR images.

9.     Figure 4: Comparison (a) vs (b): On my screen (a) seems to have a better CNR than (b) possibly due to some movement artefacts in (b).

10.  Figure 4: Comparison (c) vs (d): I agree that CE is slightly better visible in (d). But homogeneity and contrast seem to be better in (c). Could it be that the AIR coils introduced FA variations, e.g. in the cerebellum the effect seems to be most pronounced.

11.  Fig. 5: This is a tricky region to compare ghosting artefacts in sequential measurements. Could it not be that different motion caused different ghosting to occur. A hint might be the blurrier representation of the intervertebral discs?

Author Response

Reviewer 3

Authors response to reviewer comments

The authors would like to thank the reviewer for their insightful review. Listed below is a response to each comment that we believe successfully and succinctly addresses each comment / concern. In addition, the response to all 3 reviewer’s comments in aggregate have further strengthened the work to the point that it is acceptable for publication.

For the reviewer’s convenience the review has been reproduced below with the response to each comment by the authors shown in red.

Reviewer 3 comments

Open Review

English language and style

( ) Extensive editing of English language and style required
( ) Moderate English changes required
(x) English language and style are fine/minor spell check required
( ) I don't feel qualified to judge about the English language and style

Yes

Can be improved

Must be improved

Not applicable

Does the introduction provide sufficient background and include all relevant references?

( )

(x)

( )

( )

Are all the cited references relevant to the research?

( )

(x)

( )

( )

Is the research design appropriate?

( )

(x)

( )

( )

Are the methods adequately described?

( )

( )

(x)

( )

Are the results clearly presented?

( )

( )

(x)

( )

Are the conclusions supported by the results?

( )

(x)

( )

( )

Comments and Suggestions for Authors

In the presented manuscript the authors compare different coil systems for imaging the head at a field strength of 3T with the specific requirement that the posture of the patient be adopted for RT treatments. The comparison is carried out using phantom and patient measurements and evaluated using SNR and image uniformity as the major figure of merit.

General comments

  1. The original ISMRM abstract detailing the coil configuration is cited. However, since then a number of manuscripts have evaluated the performance of the AIR coils for different body regions including head imaging, e.g. DOI: 10.2214/AJR.20.22812 to mention just one. This also compares SNR in the head vs. commercially available head coils and the authors need to clearly point out the novelty of their investigation.

This work is the first to our knowledge to describe the testing and evaluation of a commercial prototype designed for the specific task of MR imaging of radiation therapy patients for treatment simulation. In addition, the work describes the use of this coil for the unique and extremely challenging situation of imaging of RT head and neck patients at 3T.

To address the concerns raised by the reviewer and to provide a review of articles published related to AIR coils, the following paragraph has been added to the discussion (See Page 19, line 452):

As the AIR coil technology outlined in this work becomes more widely available, research into and clinical use are expected to grow. For example, Cogswell et. al. [25] described the comparison of a custom built 16-channel ‘balaclava’ head coil to an 8-channel and 32-channel conventional RF head coil in which the results demonstrate that improved SNR can be achieved when these elements closely conform to the patient’s head but that this improvement is dependent upon the number of RF coils; 16-channel AIR SNR greater than 8-channel conventional SNR, while 16-channel AIR SNR is less than the 32-channel conventional coil.  Clinically, Fukui et. al. [26] have reported that improved SNR can be achieved using a commercially available AIR coil when compared to a conventional phased array coil for liver imaging. However, the work failed to disclose the number of AIR coil elements used. In contrast, Bae et. al. [27] described the comparison of a 30-channel ‘blanket’ AIR coil to a conventional 16-channel conventional anterior array for ZTE imaging of the lung which, as expected indicated increased image quality due in part to the almost doubling in coil elements in the AIR compared to conventional coil. The unique contribution of this work is the qualitative and quantitative evaluation of a pre-clinical novel (i.e., AIR) RF coil technology specifically designed to address one of the most challenging anatomic sites for MR imaging of radiation therapy patients in treatment position, namely the head and neck. Similarities exist between the data presented in this work and that described by Cogswell et. al. [25] in that both efforts use the same RF coil technology and apply it to similar anatomies (head versus head and neck). However, this effort addresses a more complex and larger anatomic region (head and neck) with the added complexity of imaging around patient specific immobilization devices. Further, the applicability of the Cogswell design to MR simulation is limited given that the coil has not been designed for use with immobilization devices, does not include the same anatomic coverage and is unlikely to be commercialized thereby limiting its widespread accessibility and use. Finally, Cogswell et. al. [25] reported on the evaluation of the prototype AIR coil on imaging of the brain in healthy volunteers. In contrast, this work reports on the challenges encountered when imaging acutely ill cancer patients who are less cooperative, are more susceptible to induce imaging artifacts due to swallowing and breathing, have much more complex anatomy compared to the brain, and are more prone to degraded image quality due to anatomy induced susceptibility differences that are exacerbated at 3.0T compared to 1.5T.

  1. Given a suitably well-designed coil we would expect better SNR ratios if the receiver arrays are “conformally” arranged with the patient’s skin (of have less distance to the patient than possible with rigid coils). So, this result can be expected. Given that the comparison with the commercial coil uses some flexible arrays as well, the results are somewhat confounding. How much SNR is gained from just placing the AIR coils closer to the patient and how much of the SNR gain is due to other factors like number of elements.

The Discussion (see Page 19, line 452) includes new text that describe three studies, two clinical and one research involving the comparison of AIR coils to conventional ones and describes the effects of moving an RF coil closer to the anatomy of interest and the resultant increase in SNR achieved. The discussion also describes the inherent properties of the AIR coil technology based on the previously published work in reference 19. The text has also been modified to identify that SNR in parallel imaging is increased due to decreased g-factors (see page 21, lines 522 - 526).

  1. One more interest that is of practical interest (at least for everybody doing MR examinations) is the question of patient preparation. While I think that we have a mature workflow using “conventional” coils (rigid and semi-rigid) it would be interesting to hear about the experience of the authors about using the “cloth-like” AIR coils on patients.

A great question. However, given the anecdotal nature of the observation that speed of application and patient comfort were improved, it is difficult to make definitive statements without appropriate sample sizing and statistical testing. To address this comment, we have modified the second paragraph of the discussion to read as follows (Page 19, line 437):

While not quantified in this study, the RT AIR coil set and related configurations provide improved ergonomics in terms of coil placement and patient comfort. This is validated by the fact that all patients and volunteers were successfully imaged without failures due to fatigue or discomfort. In addition, the AIR coils were placed by the MR technologists without supervision as they saw fit and therefore not optimized in terms of image quality. This is particularly relevant for the anatomic site studied – the head and neck – which involve the use of tightly fitting thermoplastic immobilization masks. In this context, the AIR coils were extremely forgiving and provided the most flexibility in terms of adaptation to individual body habitus.   By contrast and as seen in Figures 1(b) and (d), two ‘paddle’ coils are placed directly onto the face and mask of the individual limiting their vision, breathing, and enhancing or inducing claustrophobic sensations. This is further exacerbated by the imaging times for these sessions that can last between 30 and 50 minutes depending on the site and type of disease. Increased anxiety has the unwanted potential of increasing patient movement, both voluntary and involuntary, resulting in motion induced degradation of image quality. The ability to quickly apply surface coils such as the RT AIR set reduce imaging set up time and the potential for patient motion while the placement of portals for both the eyes and mouth (Figure 1(c)) improves patient comfort.

Specific comments

  1. The authors frequently talk about “patients undergoing MR simulation”. My apologies, but I am not familiar with the term nor can I guess its meaning.

See page 2, line 47.

In practice, a limitation of MR RT treatment simulation, i.e., the acquisition and use of MR imaging data to design and optimize radiation therapy treatments has been the ability to obtain high quality and reproducible MR images in treatment position.

  1. Introduction, lines 59-60: I agree that commercial coils are not as flexible as AIR coils but they can be suitably arranged around the patient’s head – albeit maybe with a lower number of receiver numbers. Maybe this could be rephrased for clarity.

See page 2, line 63.

Of these, the brain represents the most amenable site for the placement of smaller flexible arrays due to its roughly spherical shape but still presents with unique challenges in terms of reproducible image quality due to their semi rigid construction and overall thickness.

  1. Methods and Materials, lines 109-110: Yes, one would need to modify the coil files to adopt the system for an alternative coil configuration (be aware that be doing so the MR system uses FDA approval). However, the interesting question here would be how the signal from the different coil systems and individual elements is combined for image calculation.

See page 4, line 124.

Common to all coil sets and coil configurations was the method in which individual coil signals were combined to produce the final composite image and involved calculation of the square root of the sum of the squared signal from each coil element activated by the individual coil configuration file. Combination of individual coil images was thus independent of the type of coil element.

  1. Methods and Materials, lines 130-135: Could the authors provide details about the liquid used to fill the phantom and its composition?

See page 5, line 133.

The PVC / softener combination produced a soft, solid material that did not evaporate over time or result in ‘swirling’ imaging artifacts due to motion seen when fluids are used as signal generating materials.

  1. Materials and Methods, lines 146-147 and 167-178: Actually, the referenced NEMA standard does provide multiple methods to calculate SNR from MR images. Form the description provided it is not clear which one was used. I can only guess that “two sequential” measurements imply a measurement without the phantom to get the SD of the noise pixels and a subsequent second measurement with the phantom for the mean signal intensity? How are the 3D maps of SNR obtained?

Page 5, line 148 modified and reads:

Sequential acquisitions were obtained to reconstruct SNR data in accordance with the method proposed by the National Electrical Manufacturers Association (NEMA) standards Publication MS 6-2008 [21] and described in the following paragraph.

  1. Results: Table 2 seems to indicate some performance benefits for the AIR coil configurations. However, Fig. 3b shows better per slice SNR performance for the RT suite configuration. Please explain.

Table 2 lists the statistical test results for the SNR and uniformity data when comparing the RT_SUITE and RT_AIR coil configurations. The data indicates that in general the RT_AIR coil outperforms the RT_SUITE. Figure 3 shows the volume SNR, per slice SNR and per slice uniformity for the RT_SUITE and RT_AIR (coil configuration AF). Specifically Figure 3(b) shows that the SNR for the RT_AIR coil (blue line) is greater than the RT_SUITE coil (red line). This is supported by the Kruskal-Wallis test which showed that the AF coil configuration was statistically larger than the RT_SUITE coil (214 vs 98).

  1. Results: Figure 2c (assuming white means the region of uniformity) shows higher uniformity for the AIR coil configurations. This does not seem to be backed up by Figure 3(c) where (except for a small range of slices) the RT suite configurations performs better. Please explain.

Figure 2 has been reformatted to show the comparison SNR (a) and uniformity images(b) in portrait instead of the previous landscape format. The uniformity images are binary representations of the SNR figures (a) in which those pixels that are within 20% of the global SNR mean are set to 1 while everything else is set to 0. This is the same data that is shown in Figure 3(c). Inspection of this figure shows that the per slice uniformity measures are close but that the RT AIR (blue) coil uniformity values are higher than the RT_SUITE (red). Figure 3(b) reflects these differences but also highlight the fact that these differences are subtle and spatially variant further supporting the challenge of providing uniform MR signal across the entire imaging volume using flexible multi element RF coils such as the RT_SUITE and AIR coils described in this work.

  1. Results Figure 2(a); Please add scales to the SNR images.

Scale has been added to Figure 2(a).

  1. Figure 4: Comparison (a) vs (b): On my screen (a) seems to have a better CNR than (b) possibly due to some movement artefacts in (b).

The text has been modified to read (Page 12, line 300):

Figure 4(b) also shows improved depiction of the small enhancing nodule adjacent to the arrow tip that is not as clearly depicted in 4(a) due to increased SNR despite the increased enhancement of pulsatile flow anterior to the resection cavity and slight non-uniformity across the brain.

Figure 4: Comparison (c) vs (d): I agree that CE is slightly better visible in (d). But homogeneity and contrast seem to be better in (c). Could it be that the AIR coils introduced FA variations, e.g. in the cerebellum the effect seems to be most pronounced?

The text has been modified to read (Page 12, line 305):

Figure 4(d) also exhibits a slight loss of image quality in the cerebellum and cervical spine, the source of which is most likely due to swallowing and motion artifact. However, despite these artifacts and the loss of contrast in the cerebellum, the AIR coil system was able to capture the anatomy within and around the surgical cavity – the anatomic region in question.

  1. 5: This is a tricky region to compare ghosting artefacts in sequential measurements. Could it not be that different motion caused different ghosting to occur. A hint might be the blurrier representation of the intervertebral discs?

The authors agree with the reviewer that this is a challenging region to compare artifacts. The authors also agree that there is swallowing motion present inferiorly in (b) but despite this image quality is improved in (b) compared to (a). The text has been modified to clarify this question (see Page 14, line 326…).

Round 2

Reviewer 1 Report

The paper was improved

1. The authors replied to my comment: "The tables have been moved to Appendix A of the document. The authors are unclear as to how to present the results of the data listed in the tables in graphical form and ask that the reviewer provide more insight into how this could be achieved." - there is no need to represent the raw data in the article - instead you can just reffer to the ones now in appendics.

2. "The authors are not clear on what the reviewer is referring to as the manuscript was prepared using the journal’s template and includes all sections in the same order and format (Introduction, Materials and Methods, Results, Discussion, Conclusions). The authors have re-reviewed the work and believe that the manuscript is in compliance with the standards identified by the journal. The authors would like to request that the reviewer provide additional details regarding formatting inconsistencies." https://www.mdpi.com/journal/jcm/instructions